

# Behavioral flexibility and problem solving in an invasive bird

Corina J. Logan

SAGE Center for the Study of the Mind, University of California, Santa Barbara, CA, United States
Current affiliation: Department of Zoology, University of Cambridge, Cambridge, United Kingdom

## ABSTRACT

Behavioral flexibility is considered an important trait for adapting to environmental change, but it is unclear what it is, how it works, and whether it is a problem solving ability. I investigated behavioral flexibility and problem solving experimentally in great-tailed grackles, an invasive bird species and thus a likely candidate for possessing behavioral flexibility. Grackles demonstrated behavioral flexibility in two contexts, the Aesop's Fable paradigm and a color association test. Contrary to predictions, behavioral flexibility did not correlate across contexts. Four out of 6 grackles exhibited efficient problem solving abilities, but problem solving efficiency did not appear to be directly linked with behavioral flexibility. Problem solving speed also did not significantly correlate with reversal learning scores, indicating that faster learners were not the most flexible. These results reveal how little we know about behavioral flexibility, and provide an immense opportunity for future research to explore how individuals and species can use behavior to react to changing environments.

# BACKGROUND

Behavioral flexibility, defined here as the ability to change preferences when circumstances change based on learning from previous experience or using causal knowledge, is frequently implicated as a key factor involved in problem solving success and adapting behavior to changing environments (e.g., *Lefebvre et al., 1997*; *Griffin & Guez, 2014*; *Buckner, 2015*; *Chow, Lea & Leaver, 2016*). Those individuals or species that are more behaviorally flexible are predicted to learn faster and more efficiently, and rely on more learning strategies to solve problems (*Griffin & Guez, 2014*). Testing behavioral flexibility experimentally requires individuals to change their behavior in response to changes in the task.

A common way to experimentally test behavioral flexibility uses a reversal learning paradigm where individuals first learn to prefer a particular option, and then, once proficient, the reward contingencies are altered such that previously correct choices are now unsuccessful, therefore the individual must learn to change its preference (e.g., *Bond, Kamil & Balda, 2007*; *Tebbich, Sterelny & Teschke, 2010*; *Ghahremani et al., 2010*; *Buckner, 2015*). The few investigations that have examined the relationship between behavioral flexibility and problem solving abilities produced mixed results. Two studies investigating behavioral flexibility and problem solving speed found that faster learners were slower

Corresponding author
Corina J. Logan, cl417@cam.ac.uk

to reverse their preferences (song sparrows: *Boogert et al., 2011*, invasive Indian mynas: *Griffin et al., 2013*, threatened Florida scrub-jays: *Bebus et al., 2016*). Another study found no correlation between reversal learning speed and problem solving speed or ability (color and shape discrimination, spatial memory, and motor skills; spotted bowerbirds: *Isden et al., 2013*). Reversal learning speed is thought to positively correlate with inhibition: when the task changes subjects must inhibit the previously learned behavior to be able to learn the new behavior (*Manrique, Völter & Call, 2013*; *Griffin & Guez, 2014*; *Liu et al., 2016*). However, this idea is challenged by an experiment in rats that were genetically modified to increase inhibition (*Homberg et al., 2007*). Knock out rats with improved inhibition showed no difference in their reversal learning speed from non-modified rats (*Homberg et al., 2007*). This suggests that behavioral flexibility may rely more on individuals continuing to sample their environment rather than simply inhibiting a response when a behavior is no longer rewarded. This variety of contrasting results indicates how little is known about behavioral flexibility in terms of how it relates to problem solving abilities and cognition.

*Griffin & Guez (2014)* propose that behavioral flexibility is a multi-faceted trait: some aspects are measurable in problem solving tasks while other aspects are measurable in other contexts, therefore individuals might exhibit flexibility in some contexts but not others. Behavioral flexibility is usually studied in relation to problem solving speed (*Griffin et al., 2013*; *Bebus et al., 2016*), not problem solving success, and, while behavioral flexibility is also tested in non-reversal learning paradigms (e.g., multi-access box: *Auersperg et al., 2011*; *Manrique, Völter & Call, 2013*; episodic-like memory and future planning: *Clayton & Dickinson, 1998*; *Dally, Emery & Clayton, 2006*; *Raby et al., 2007*), it is generally tested in only one context per study. Therefore, our understanding of the mechanisms underlying behavioral flexibility is lacking.

To begin to address these gaps, I investigated behavioral flexibility in one of the most invasive species in North America, the great-tailed grackle (*Quiscalus mexicanus*, family Icteridae, hereafter referred to as grackles; *Peer, 2011*). Species that rapidly adapt to novel environments are presumed to require the ability to behaviorally respond to changing circumstances within the course of their lifetime (*Sol & Lefebvre, 2000*; *Sol, Timmermans & Lefebvre, 2002*; *Sol et al., 2005*; *Sol et al., 2007*), thus many invasive species are likely candidates for possessing behavioral flexibility. I investigated whether grackles are behaviorally flexible and efficient problem solvers, whether they vary in behavioral flexibility across contexts, whether flexibility correlates with problem solving ability and speed, and whether individuals that are more flexible use more learning strategies.

I tested behavioral flexibility in two contexts by measuring preferences (due to learning, attending to function, or previous experience) and then requiring individuals to change preferences after modifying the task. A color association task (context 1) involved a gold tube and a silver tube placed on the table at the same time and with one of the tubes containing hidden food. Individuals learned to associate food with first the gold tube (learning speed; Experiment 1) and then the silver tube (a modified version of reversal learning in that there was only 1 reversal; Experiment 2). I used this task to compare the speed with which grackles learn and reverse preferences compared with other species, and to examine which learning strategies grackles use to become proficient. Probability theory calls this type of problem a contextual, binary multi-armed bandit and predicts

two different learning strategies as solutions to the problem (*McInerney, 2010*). These learning strategies involve a trade off between an exploration phase and an exploitation phase. The pattern of the trade off indicates which learning strategy was used. For example, a short exploration phase before switching almost exclusively to the exploitation phase (marked by significantly more correct choices) indicates one learning strategy, whereas a long exploration phase before eventually choosing significantly more correct choices (exploitation phase) indicates a different learning strategy.

The Aesop's Fable paradigm (context 2) examines problem solving ability and involves food floating in a partially filled water tube, which is solved by inserting objects into the tube to raise the water level and bring the food within reach. It has been used to explore the cognitive abilities underlying problem solving in rooks (*Bird & Emery, 2009*), Eurasian jays (*Cheke, Bird & Clayton, 2011*), humans (*Cheke, Loissel & Clayton, 2012*), New Caledonian crows (*Taylor et al., 2011*; *Jelbert et al., 2014*; *Logan et al., 2014*), and Western scrub-jays (*Logan et al., 2016*). While great-tailed grackles are not reported to use tools (they are not listed in *Lefebvre, Nicolakakis & Boire, 2002* or in *Shumaker, Walkup & Beck, 2011*), non-tool using species have successfully participated in the Aesop's Fable tests (rooks: *Bird & Emery, 2009*, Eurasian jays: *Cheke, Bird & Clayton, 2011*, and Western scrub-jays: *Logan et al., 2016*) as well as other tool-using tests (e.g., the trap tube; rooks: *Seed et al., 2006*; *Tebbich et al., 2007*), therefore I expect grackles to be capable of performing these experiments. I compared grackle problem solving performance with previously tested species to determine whether grackles are efficient problem solvers. I modified the standard Aesop's Fable experiments to test behavioral flexibility by assessing whether they prefer to drop the more functional heavy objects in a Heavy vs. Light experiment, and whether they change these preferences in a follow up experiment where the heavy objects become non-functional (Heavy vs. Light Magic).

## GENERAL METHODS

### Ethics

This research was carried out in accordance with permits from the US Fish and Wildlife Service (scientific collecting permit number MB76700A-0), California Department of Fish and Wildlife (scientific collecting permit number SC-12306), US Geological Survey Bird Banding Laboratory (federal bird banding permit number 23872), and the Institutional Animal Care and Use Committee at the University of California Santa Barbara (IACUC protocol numbers 860 and 860.1).

### Subjects and study site

Eight wild adult great-tailed grackles (4 females and 4 males) were caught using a walk-in baited trap measuring 0.61 m high by 0.61 m wide by 1.22 m long (design from *Overington et al., 2011*). Birds were caught (and tested) in two batches: batch 1 at the Andree Clark Bird Refuge (4 birds (Tequila, Margarita, Cerveza, and Michelada) in September 2014, released in December) and batch 2 at East Beach Park (4 birds (Refresco, Horchata, Batido, and Jugo) in January 2015, released in March) in Santa Barbara, California. They were housed individually in aviaries measuring 183 cm high by 119 cm wide by 236 cm long at the University of California Santa Barbara for 2–3 months while participating in the

experiments in this study. Grackles were given water *ad libitum* and unrestricted amounts of food (Mazuri Small Bird Food) for at least 20 h per day, with their main diet being removed for up to 4 h on testing days while they participated in experiments and received peanuts or bread when successful. Grackles were aged by plumage and eye color and sexed by plumage and weight following *Pyle (2001)*. Biometrics, blood, and feathers were collected at the beginning and end of their time in the aviary. Their weights were measured at least once per month, first at the time of trapping using a balancing scale, and subsequently by placing a kitchen scale covered with food in their aviary and recording their weight when they jumped onto the scale to eat.

## Experimental set up

Apparatuses were placed on top of rolling tables (60 cm wide by 39 cm long) and rolled into each individual's aviary for testing sessions, which lasted up to approximately 20 min. If habituation to an apparatus was needed, as indicated by a bird's unwillingness to immediately approach and eat from it (this was the case for most birds with the stone dropping training apparatus), it was placed in their aviary overnight and they were fed from it. If an apparatus had parts that would allow a bird to learn how the task worked, these parts were taped over to prevent learning. If a grackle approached an apparatus and ate from it without hesitating, it was considered habituated. If re-habituation was needed, as indicated by an unwillingness to approach the apparatus after the training or experiment began, the habituation process was repeated. Color tubes were baited with peanut pieces and/or bread. Water tubes were baited with 1/16 of a peanut attached to a small piece of cork with a tie wrap for buoyancy (hereafter referred to as a peanut float). The area around the top of the water tube (the standing platform) was also sometimes baited with smaller peanut pieces and bread crumbs, and more peanut floats could be added to the inside of the water tube to encourage the bird to interact with the task. If more than one peanut float was in the tube, the bird was given the opportunity, after retrieving the first peanut float, to insert more objects into the tube to retrieve the other peanut floats. If a bird started to lose motivation (e.g., refuse to come to the table or interact with the apparatus) for participating in a task because they were unsuccessful (as in Heavy vs. Light Magic), I baited the standing platform between trials to reward their participation and keep them interested in finishing the experiment. A trial was terminated when the bird solved the task or did not interact with the apparatus for at least 1 min. All water tube experiments (3–6) consisted of 20 trials per bird and were recorded with a Nikon D5100 camera on a tripod placed inside the aviary. Experiments were given in the following order: 1, 3, 4, 5, 6, 2, which was the same for all birds (except Tequila who began, but did not complete, a sand vs. water experiment between Experiments 1 and 3). Grackles took 1–7 days to complete an experiment, which spanned the course of up to 19 days. Birds were tested with the experimenter just outside the aviary door and in full sight of the grackle. Grackles are already habituated to humans because they forage in an urban setting, often coming within a meter of the nearest humans. They quickly habituated to the testing set up and their first 5 days in the aviary were spent habituating them to a human presence just outside their aviary door.

### Experimenters

My research assistant, Luisa Bergeron, and I conducted Experiments 1–2; I conducted Experiments 3–6; and my assistants (Luisa Bergeron, Alexis Breen, Michelle Gertsvolf, Christin Palmstrom, and Linnea Palmstrom) and I conducted the stone dropping training.

### Statistical analyses
#### Color association tests

Two analyses were performed on the color association data (Experiments 1 and 2). First, a bird was considered to pass this test if it chose correctly at least 17 out of the most recent 20 trials (with a minimum of 8 or 9 correct choices out of 10 on the two most recent sets of 10; binomial test: $p = 0.003$ for 17/20). Once the bird reached proficiency using this analysis, their individual learning strategy was identified using a contextual, binary multi-armed bandit (see *McInerney, 2010* for a review). It was contextual in that the subject was allowed to make only one choice per trial, and binary because there were two options on the table, one containing a reward and the other containing no reward. I categorized grackle learning strategies by matching them to the two known approximate strategies of the contextual, binary multi-armed bandit: epsilon-first and epsilon-decreasing (*McInerney, 2010*). The following equations refer to the different phases involved in each strategy:

Equation 1 (exploration phase): $\epsilon N$

Equation 2 (exploitation phase): $(1 - \epsilon)N$

$N$ is the number of trials given, and epsilon, $\epsilon$, represents the subject's uncertainty about the location of the reward, starting at complete uncertainty ($\epsilon = 1$) at the beginning of the experiment and decreasing rapidly as individuals gain experience with the task (exploration phase where the rewarded color is chosen below or at chance levels) and switch to the exploitative phase (the rewarded color is chosen significantly above chance levels). Because the grackles needed to learn the rules of the task, they necessarily had an exploration phase. The epsilon-first strategy involves an exploration phase followed by an entirely exploitative phase. The optimal strategy overall would be to explore one color in the first trial and the other color in the second trial, and then switch to an exploitative strategy (choose the rewarded color significantly above chance levels). In this case there would be no pattern in the choices in the exploration phase because it would consist of sampling each color only once. In the epsilon-decreasing strategy, birds would start by making some incorrect choices and then increase their choice of gold gradually as their uncertainty decreases until they choose the rewarded color significantly above chance levels. In this case, a linear pattern emerges during the exploration phase.

To determine whether faster learners are also more flexible, I used a Spearman's rank correlation test to examine whether learning speed (number of trials to learn a preference) positively correlated with reversal learning scores (number of trials to reverse a preference minus the number of trials to learn the preference) in the color association test.

### Aesop's Fable tests

To make the water tube experimental results comparable with previous studies, I used two-tailed binomial tests to determine whether each grackle chose particular objects or tubes at random chance (null hypothesis: $p \geq 0.05$) or significantly above chance (alternative hypothesis: $p < 0.05$). The Bonferroni–Holm correction was applied to $p$-values within each experiment to correct for an increase in false positive results that could arise from conducting multiple tests on the same dataset.

Generalized linear mixed models (GLMMs) were used to determine whether birds preferred particular objects or tubes (response variable: correct/more correct choice or incorrect/less correct choice) in a water tube experiment and whether the trial number or bird influenced choices (explanatory variables: experiment, trial number, bird), and to control for the non-independence of multiple choices per trial (random factor: choice number). I used minimal belief priors ($V = 1$, nu $= 0$) and fixed the variance component to one (fix $= 1$) because the measurement error variance was known, as is standard when choices are binary (*Hadfield, 2010*). I ensured that the Markov chain for this test model converged by manipulating the number of iterations (nitt $= 150,000$ for the null model, nitt $= 600,000$ for the test model), the number of iterations that must occur before samples are stored (burnin $= 30,000$), and the intervals the Markov chain stores (thin $= 300$) until successive samples were independent as indicated by low ($<0.1$) correlations (autocorr function, MCMCglmm package: *Hadfield, 2014a*; *Hadfield, 2014b*) and there were no trends when visually inspecting the time series of the Markov chain (function: plot(testmodel$Sol); *Hadfield, 2014a*; *Hadfield, 2014b*). I compared this test model to a null model where I removed all explanatory factors and set it to 1.

I determined whether the test model was likely given the data, relative to the null model, by using Akaike weights (range: 0–1, all model weights sum to 1; *Akaike, 1981*; Weights function, MuMIn package: *Bates, Maechler & Bolker, 2011*). The Akaike weight indicates the "relative likelihood of the model given the data" (*Burnham & Anderson, 2002*, p. xxiii) and models with Akaike weights greater than 0.89 are considered reliable models because they are highly likely given the data (*Burnham & Anderson, 2002*). The test model was highly likely given the data (Akaike weight $= 1.00$) and the null model was not (Akaike weight $= 3.4\text{e}{-30}$). To investigate the potential effects of season or order of testing, I conducted a GLMM to determine whether the batch to which the bird belonged (explanatory variable: batch $= 1$ or 2) influenced their test performance (response variable: correct or incorrect choice) while controlling for the non-independence of multiple choices per trial (random factor: choice number). The null model was highly likely given the data (Akaike weight $= 0.94$), while the batch model was not (Akaike weight $= 0.06$), indicating that batch did not influence test performance. GLMMs were carried out in R v3.2.1 (*R Core Team, 2016*) using the MCMCglmm function (MCMCglmm package, *Hadfield, 2014a*) with a binomial distribution (called categorical in MCMCglmm) and logit link.

### Cross-context analysis

To determine whether those grackles that were more behaviorally flexible in the water tube context (yes or no) were also more behaviorally flexible in the color association context
(the number of trials to meet criterion in Experiment 2 minus the number of trials to meet criterion in Experiment 1), I used a Spearman's rank correlation test. Margarita and Cerveza were included, even though their attraction to the magnet in Experiment 4 biased their results, to increase the sample size of individuals that participated in both contexts to 5.

### Data availability
The data are available at the KNB Data Repository: https://knb.ecoinformatics.org/#view/doi:10.5063/F1319SVV (*Logan, 2016*).

### Video
Watch video clips showing examples of each experiment at: https://youtu.be/fdCJGwvaDsk.

## CONTEXT 1: COLOR ASSOCIATION TESTS

### Experiment 1: color association test (learning speed)
#### Experiment 1: methods
To assess how many trials it takes a grackle to form an association between food and color, they were given a gold and a silver tube with food (peanut pieces or bread) always hidden in the gold tube (*Logan et al., 2014*; *Logan et al., 2016*). Grackles were first trained on a blue tube where they learned to search for hidden food. Each color tube set up consisted of a PVC tube (outer diameter 26 mm, inner diameter 19 mm) mounted on two pieces of plywood glued together at a right angle (whole apparatus measuring 50 mm wide by 50 mm tall by 67 mm deep). Each tube was placed at opposite ends of a table with the tube openings facing the side walls so the bird could not see which tube contained the food. Tubes were pseudorandomized for side and the left tube was always placed first, followed by the right to avoid behavioral cueing. Pseudorandomization consisted of alternating location for the first two trials of a session and then keeping the same color on the same side for at most 2 consecutive trials thereafter. Each trial consisted of placing the tubes on the table, and then the bird had the opportunity to choose one tube by looking into it (and eating from it if it chose the gold tube). Once the bird chose, the trial ended by removing the tubes.

#### Experiment 1: results
All grackles learned to associate the gold tube with a food reward by reaching criterion (at least 17/20 correct trials) in 20–40 trials (Table 1). Using the binary multi-armed bandit analysis to assess which learning strategies birds used, Refresco used the epsilon-first strategy because he first explored (i.e., made unsuccessful and/or successful choices) and then exploited (i.e., was successful) every trial thereafter: he explored in his first trial (he failed by choosing silver) and then always chose gold after that (Fig. 1). The rest of the grackles used the epsilon-decreasing strategy by exploring more at the beginning and gradually increasing their success until they performed above chance levels by the end of the experiment (Fig. 1). Horchata and Jugo had exceptions to this strategy: Horchata started a second exploration phase at the end of her experiment, and Jugo's pattern of exploration did not linearly increase in his first several trials (though it did thereafter). In

**Table 1  Color association results.** The number of trials needed to reach proficiency in Experiment 1 (learning speed), and Experiment 2 (reversal learning speed), and the number of trials needed to pass the Experiment 1 refresher (*, did not pass the refresher in 10 trials).

| Bird | Sex | Experiment 1: learning speed | Experiment 2: reversal learning speed | Experiment 1 refresher |
| --- | --- | --- | --- | --- |
| Tequila | M | 30 | 100* | 30 |
| Margarita | F | 30 | 100 | 10 |
| Cerveza | F | 30 | 90 | 10 |
| Michelada | F | 40 | 70 | 10 |
| Horchata | F | 30 | 130* | 50 |
| Refresco | M | 20 | 70* | 80 |
| Batido | M | 30 | Incomplete* | 30 |
| Jugo | M | 40 | 80 | 10 |

the first part of Jugo's experiment, he did not appear to follow any particular rules during his learning phase such as 'always choose the left side' or 'always alternate sides', therefore it is unknown how this part of his exploration phase should be classified according to probability theory.

### Experiment 1: discussion

Grackles were fast to learn an initial preference in the color association task (average 31 trials). Their performance is similar to Western scrub-jays (*Logan et al., 2016*), 3 species of Darwin's finches (*Tebbich, Sterelny & Teschke, 2010*), and pigeons (*Lissek, Diekamp & Güntürkün, 2002*) who learned in an average of between 40 and 56 trials using a similar experimental design and passing criterion. These species are faster than Pinyon jays, Clark's nutcrackers, a different group of Western scrub-jays (*Bond, Kamil & Balda, 2007*), and Indian mynas (*Griffin et al., 2013*) who learned on average between 122 and 280 trials and also had a similar passing criterion and experimental design. The Indian myna's design was different, however they are included for comparison because they are one of the only other species that has been tested in a similar behavioral flexibility-problem solving context. If grackles trade off learning speed for inhibition (implicated in behavioral flexibility; *Manrique, Völter & Call, 2013*; *Griffin & Guez, 2014*; *Liu et al., 2016*) or continuing to sample their environment, as Indian mynas and Florida scrub jays appear to (*Griffin et al. (2013)*; *Bebus et al. (2016)*), then those grackles that are faster learners will be the least flexible (e.g., the slowest to reverse their initially learned preference). However, if these traits are independent, then I expect grackle learning speed not to correlate with flexibility as in spotted bowerbirds (*Isden et al., 2013*).

### Experiment 2: color association reversal (learning speed)
### Experiment 2: methods

The methods were the same as in the color association test (Experiment 1), except the food was always placed in the silver tube rather than the gold tube, thus forcing the bird to reverse their preference to consistently obtain the food. Because many other experiments (3–6)

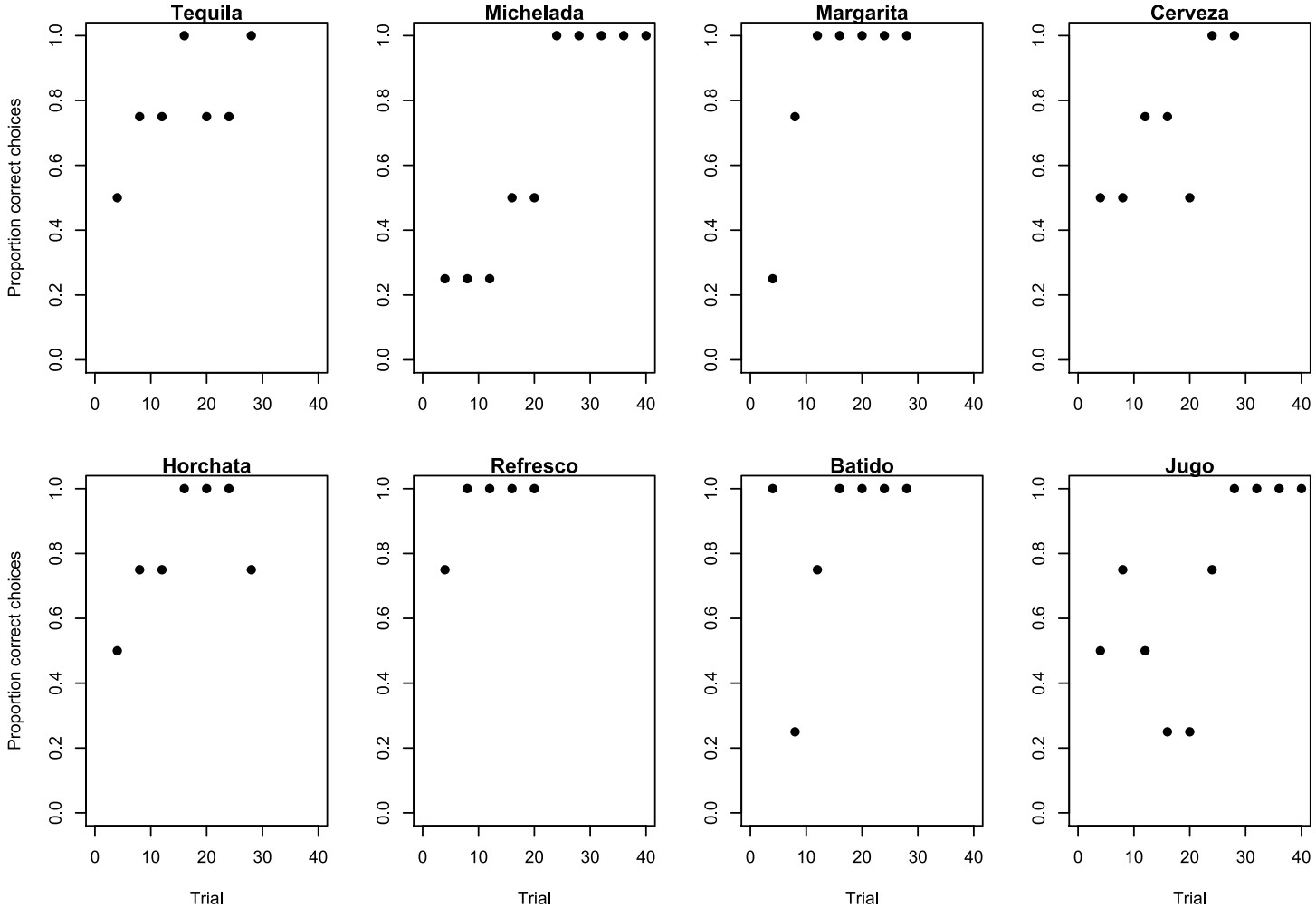

**Figure 1   Learning strategies used during Experiment 1.** Learning strategies employed by grackles when learning to associate the gold tube with food as shown by the proportion of correct choices (non-overlapping sliding window of 4-trial bins) across the number of trials required to reach the criterion of 17/20 correct choices.

occurred between Experiments 1 and 2, I first checked whether the grackles remembered Experiment 1 before moving them to Experiment 2. If they were successful in 9 or 10 out of their first 10 trials, indicating that they remembered that the food was always in the gold tube, then they moved onto reversal learning with the food always in the silver tube. If they were not successful in their first 10 trials, then they were given a refresher on Experiment 1 until they re-passed the original criterion before moving onto reversal learning.

### Experiment 2: results

For their Experiment 1 refresher, Margarita, Cerveza, Michelada, and Jugo remembered that food was always in the gold tube because they passed in their first 10 trials (Table 1 and Fig. 2). Tequila, Horchata, Refresco, and Batido needed to re-achieve proficiency on Experiment 1, requiring 30–80 trials before moving onto Experiment 2 (Table 1). Their re-learning patterns followed the epsilon-decreasing strategy that all birds used before,

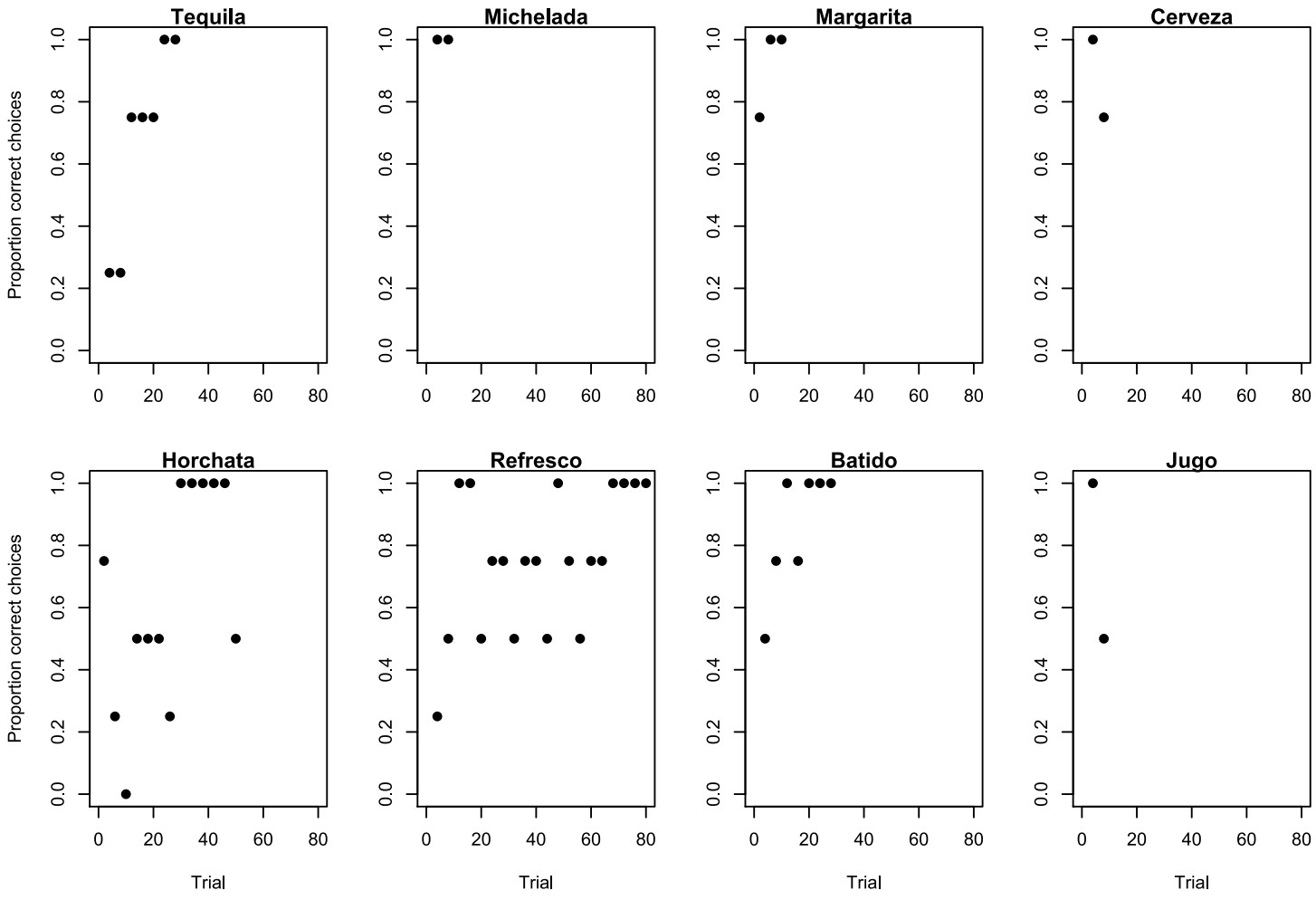

**Figure 2** **Learning strategies used during the Experiment 1 refresher.** Learning strategies employed by grackles when checking whether they re-member that the gold tube contained the food as shown by the proportion of correct choices (non-overlapping sliding window of 4-trial bins) across the number of trials required to reach the criterion of 17/20 correct choices.

except for Refresco who used the epsilon-first strategy the first time and switched to the epsilon-decreasing strategy for the refresher (Fig. 2).

Most grackles reversed their color association, indicating they were behaviorally flexible. Seven out of 8 grackles met the reversal learning success criterion (17 correct choices out of the most recent 20 trials) in 70–130 trials (Table 1), but Batido stopped participating before reaching criterion (Fig. 3). All birds used the epsilon-decreasing strategy, but they were slower to learn to reverse their previously learned preference than they were to initially learn the preference, and many continued to explore throughout the experiment (Fig. 3).

Faster learners were not less flexible. In the color association test, learning speed (number of trials to learn a preference) did not significantly correlate with reversal learning scores (number of trials to reverse a preference minus the number of trials to learn the preference; Spearman's rank correlation test: $S = 84.14$, $p = 0.25$, rho $= -0.50$, $n = 7$).

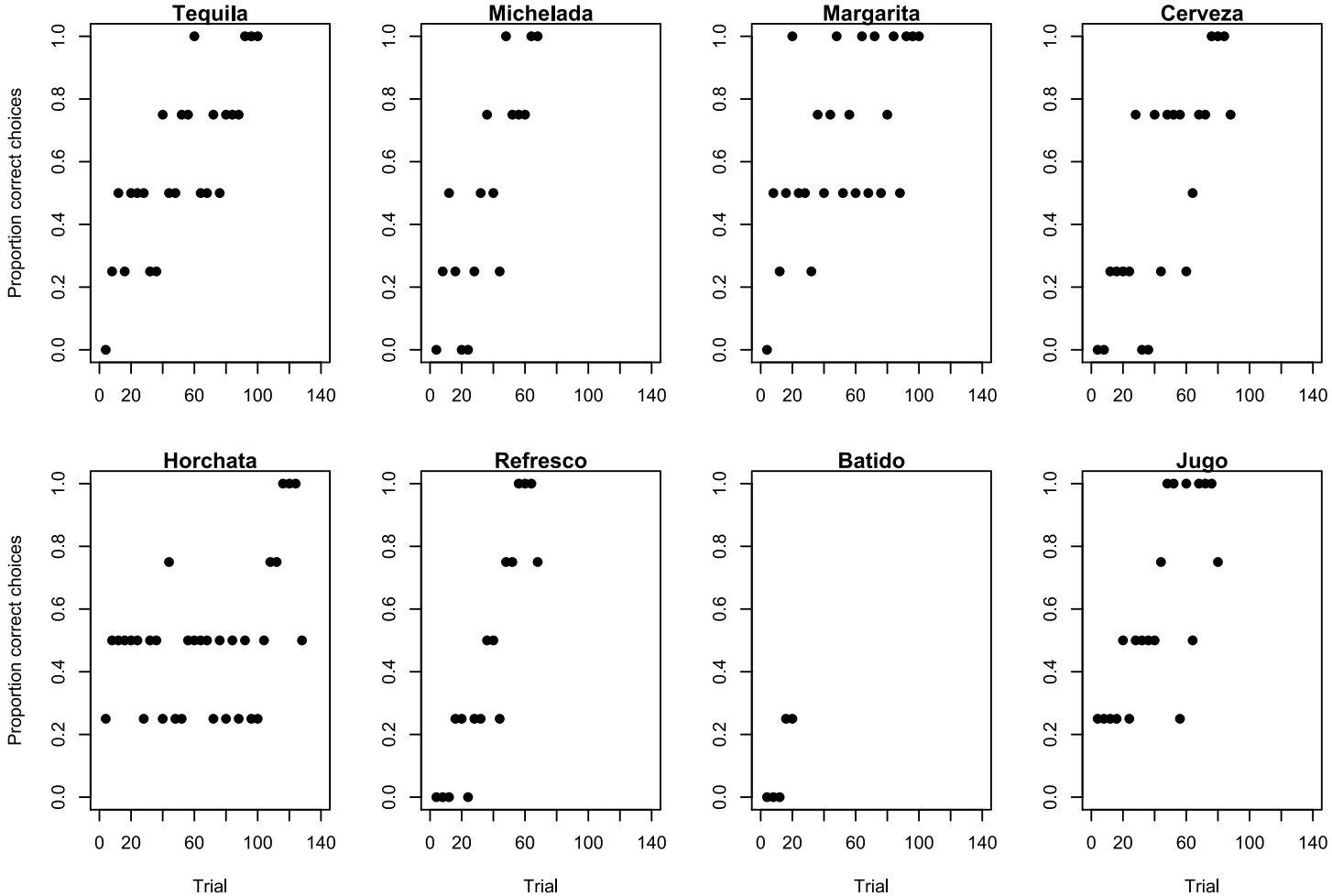

**Figure 3 Learning strategies used during reversal learning in Experiment 2.** Learning strategies employed by grackles when learning to associate the silver tube with food, rather than their previously learned association between the gold tube and food, as shown by the proportion of correct choices (non-overlapping sliding window of 4-trial bins) across the number of trials required to reach the criterion of 17/20 correct choices.

### Experiment 2: discussion

Grackles demonstrated behavioral flexibility in the color association task by quickly reversing their initially learned preference (average 91 trials). Their performance was similar to 3 species of Darwin's finches who reversed in an average of 76–95 trials (*Tebbich, Sterelny & Teschke, 2010*). Darwin's finches and grackles reversed more quickly than pigeons (*Lissek, Diekamp & Güntürkün, 2002*), Pinyon jays, Clark's nutcrackers, Western scrub-jays (*Bond, Kamil & Balda, 2007*), and Indian mynas (*Griffin et al., 2013*) who learned on average between 142 and 380 trials.

Faster learners were not less flexible (e.g., slower to reverse a preference), which is not consistent with the prediction that learning speed trades off with inhibition or continuing to sample the environment, which was found in Indian mynas and Florida scrub-jays (*Manrique, Völter & Call, 2013*; *Griffin et al., 2013*; *Griffin & Guez, 2014*; *Bebus et al., 2016*; *Liu et al., 2016*). There was no correlation between grackle learning speed and behavioral

flexibility, similar to results found in spotted bowerbirds that were tested in the wild using a different experimental design and passing criterion (6 consecutive correct choices before trying an incorrect choice in 2 consecutive sessions; *Isden et al., 2013*). Regardless of methodological differences, such unpredictable variation in behavioral flexibility across species suggests that the underlying mechanisms are poorly understood and require further investigation.

## CONTEXT 2: AESOP'S FABLE TESTS

### Context 2: general methods and results
#### *Spontaneous stone dropping*
**Methods:** Birds were given two sequential 5 min trials with the stone dropping training apparatus and two stones to see whether they would spontaneously drop stones down tubes. The stone dropping training apparatus was a clear acrylic box with a tube on top. The box contained out of reach food on top of a platform that was obtainable by dropping a stone into the top of the tube, which, when contacting the platform, forced the magnet holding it up to release the platform (design as in *Bird & Emery, 2009* with the following tube dimensions: 90 mm tall, outer diameter = 50 mm, inner diameter = 37 or 44 mm). The food then fell from the platform to the table. At the end of the first 5 min trial, the stones were moved to different locations on the table and on the wooden blocks. The blocks made it easier to access the top of the tube.

**Results:** No grackle spontaneously dropped stones down the tube of the platform apparatus, indicating that this was not a behavior that was easily innovated. Therefore, they all underwent stone dropping training to allow them to participate in Experiments 3–6.

#### *Stone dropping training*
**Methods:** Those birds that did not spontaneously drop stones down the tube on the stone dropping training apparatus were trained to push or drop stones down tubes using this same apparatus (Fig. 4). Birds were given two stones and went from accidentally dropping stones down the tube as they pulled at food under the stones, which were balanced on the edge of the tube opening, to pushing or dropping stones into the tube from anywhere near the apparatus. Once the bird proficiently pushed or dropped stones into the apparatus 30 consecutive times, they moved on to obtain their reachable distance on a water tube.

Stone pushing/dropping proficiency was defined as consistently directing the stone to tube opening from anywhere on the standing platform at the top of the apparatus. Not all motions had to be in the direction of the tube opening because some grackles preferred to move the stone to a particular location on the standing platform (which may initially be in the opposite direction from the tube) and push or drop it in from there, or push the stone in shorter, angular strokes. It was permissible for a bird to throw one of the stones off the side of the apparatus (which occurred sporadically in training and experiments) as long as they proficiently put the other stone in the tube.

The training procedure was modified from *Logan et al. (2014)* to allow stone pushing from a clear cast acrylic standing platform placed on top of the tube rather than stone dropping by picking up the stone from the table and putting it into the tube without a

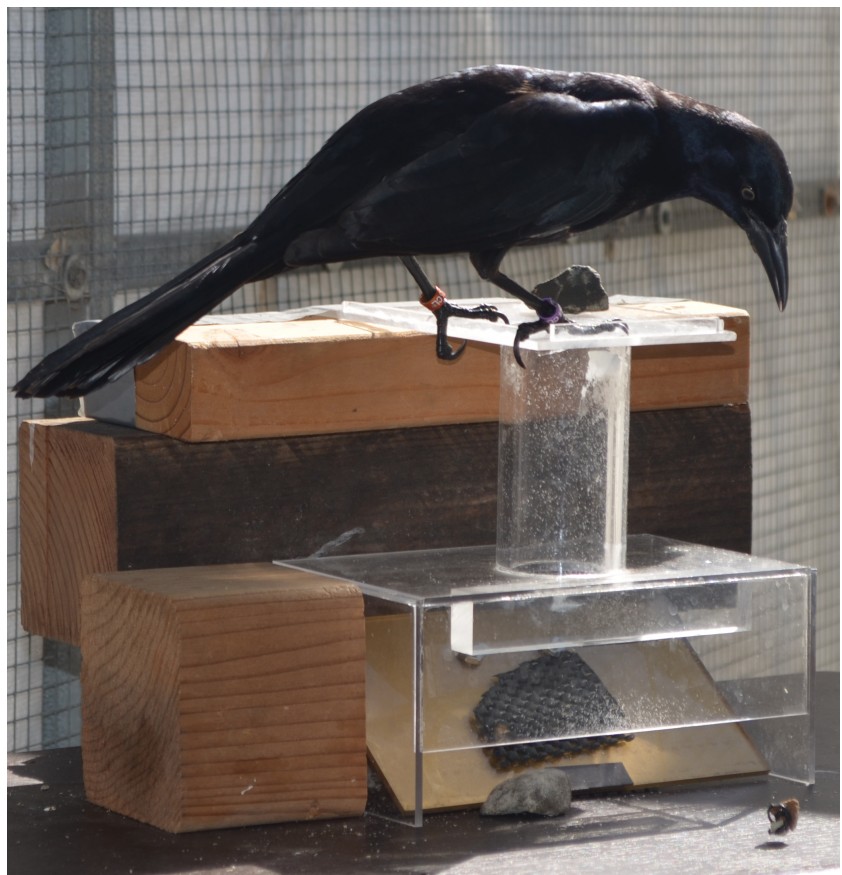

**Figure 4** **Stone dropping training using a platform apparatus.** Batido inserted a stone into the stone dropping training apparatus, which collapsed the platform and the peanut float fell out onto the table.

standing platform (Fig. 4). The modification was necessary because grackles seem to form associations between the stones and the top of the tube, the stones and the table where the food comes out, and the stones falling in one direction only: down. When I placed the stones below the level of the top of the tube to try to train them to pick the stones up and put them in the top of the tube, the grackles took the stones and dropped them off the side of the apparatus or table, often placing them on the table and then looking at where the platform should have fallen open. Placing the standing platform at the tops of the water tubes for the experiments was implemented to mitigate this limitation. Once this change was made, it was no longer necessary to train the grackles to pick up and drop the stones because pushing them into the tube sufficed and required less training.

Similar to Western scrub-jays (*Logan et al., 2016*), the grackles inserted objects while standing at the top of the tube rather than standing on the ground. The different standing position should not influence their perception of the objects as they were inserted into the tube because their heads were always over the top of the tube at the time of insertion, regardless of where they were standing.

**Results:** Most grackles learned to push stones into a tube on the platform apparatus in 135–362 trials (Table 2), however Michelada showed a neophobic reaction (refused to

**Table 2 Performance per bird in the water tube experiments (3–6).** The number of stone dropping training trials needed to reach proficiency, and the number of correct choices out of the total number of choices made and *p*-values from Bonferroni-Holm corrected (within experiment) binomial tests for each experiment (−, was not given this experiment).

| Bird (color rings) | Sex | Stone drop training trials | Heavy vs. light | Heavy vs. light magic | Wide vs. narrow equal |
|---|---|---|---|---|---|
| Tequila (YP) | M | 192 push/233 drop | 33/43 0.003 (heavy) | 19/30 0.60 | – |
| Margarita (PB) | F | 362 | 41/49 0.00001 (heavy) | 17/56 0.02 (heavy) | – |
| Cerveza (BO) | F | 252 | 36/55 0.06 | 10/39 0.02 (heavy) | 1.00 |
| Michelada (OR) | F | – | – | – | – |
| Batido (OP) | M | 179 | 38/51 0.002 (heavy) | 28/37 0.02 (heavy) | 1.00 |
| Horchata (GR) | F | 135 | 18/32 0.60 | 16/32 1.00 | – |
| Refresco (PY) | M | 204 | 46/67 0.009 (heavy) | 17/35 1.00 | 1.00 |
| Jugo (RB) | M | – | – | – | – |

Notes.

Tequila was the first bird tested and I did not realise until after I trained him to pick up and drop the stones into the tube that I wanted to only train the other birds to push the stones into the tube to save training time. Therefore, the trial numbers for the other birds refer to proficiency to push objects into the tube, not pick up and drop them.
Y, Yellow; P, Purple; B, Blue; O, Orange; R, Red; G, Green.

approach and interact with the apparatus even after repeated re-habituation attempts) to the stone falling down the tube and did not habituate to this event, and Jugo learned too slowly to become proficient by the time he needed to be released. Therefore, Michelada and Jugo were excluded from the stone dropping experiments.

## Reachable distance

To determine how high to set the water levels in the Aesop's Fable experiments, a bird's reachable distance was obtained. Food was placed on cotton inside a resealable plastic bag, which was stuffed inside a standard water tube (a clear acrylic tube (170 mm tall, outer diameter = 51 mm, inner diameter = 38 mm) super glued to a clear acrylic base (300 × 300 × 3 mm)) to obtain the reachable distance without giving the bird experience with water. The food was first placed within reach and then lowered into the tube in 1 cm increments until the bird could not reach it. The lowest height the bird could still reach was considered its reachable distance and water levels in subsequent experiments were set to allow the desired number of objects to bring the food within reach.

## An additional experiment that only Tequila began: water vs. sand

**Methods:** To determine whether grackles can discriminate between functional and non-functional substrates, two standard water tubes were placed on the table: one partially filled with water (functional) and the other partially filled with sand (non-functional) to equal

levels (pseudorandomized for side; similar to *Jelbert et al., 2014*; *Logan et al., 2014*; *Logan et al., 2016*). Before the experiment began, birds were habituated to the water and sand tubes for 10 trials by taping over the openings at the top and placing food (peanut pieces) on the tops and at the bases of each tube (there was no food inside the tubes). A trial continued until the bird ate all four food pieces. The first tube from which food was taken was recorded and used as an indicator of a potential preference. Preferences were discouraged by placing relatively more bait on the non-preferred tube in the next trial. During the 20-trial experiment, four stones were placed in pairs on the standing platforms at the top of each tube, and birds could insert them into the functional water tube or non-functional sand-tube.

**Results:** Tequila participated in two trials, inserting 1 stone into the water tube in trial 1, and 1 stone into first the water tube and then 1 stone in the sand tube in trial 2. However, he started to refuse to participate in stone dropping all together, therefore I eliminated this experiment and implemented a water tube proficiency assessment to re-motivate him to participate in stone dropping experiments.

## Water tube proficiency assessment

**Methods:** Upon the implementation of the water tube proficiency assessment for Tequila after the Water vs. Sand experiment, I required this assessment for all of the other birds to ensure a more similar level of water tube experience and to ensure they were able to transfer their stone dropping skills from a platform apparatus to a water tube context. This was likely not necessary for the other grackles though because Tequila transferred his stone dropping skills directly from the platform apparatus to a water tube. Grackles were given a standard tube partially filled with water with a peanut float and four stones (9–14 g, each displaces 5–6 mm water), which they could drop into the tube to raise the water level and consequently reach the food. Once a bird accomplished 30 consecutive proficient trials, they moved on to Experiment 3. Proficiency was defined as in the stone dropping training section above.

**Results:** Most grackles immediately applied their stone dropping skills to a water tube context as indicated by their proficiency on their first trial (Cerveza, Margarita, Refresco, and Batido). Horchata was proficient by her second trial. After Tequila's refusal to insert stones into tubes following his Water vs. Sand experiment, he needed 76 trials of water tube proficiency assessment to complete stone insertion proficiency again.

## Accidental object insertions

Because objects were placed near the top of the tube to allow birds to push objects into the tube, it was also possible to accidentally push or kick an object into the tube. Accidental insertions were noted (see Tables S1–S3) and included in analyses because birds could learn about the affordances of the task if an object fell into the water, regardless of whether it was chosen or accidental. Some trials were allowed to consist of only an accidental insertion or insertions because the bird was losing motivation (e.g., refusing to come to the table or interact with the apparatus) and would not have finished the trial otherwise. Including accidental insertions in analyses errs on the conservative side because throwing these data out removes the ability to account for learning.

### Experiment 3: heavy vs. light (problem solving ability)

*Experiment 3: methods*

I modified the Aesop's Fable paradigm to test behavioral flexibility by requiring birds to change preferences using four experiments involving two preference changes, similar to reversal learning experiments. In Experiment 3 (Heavy vs. Light), grackles were given heavy and light objects with the former being twice as functional as the latter, therefore grackles should prefer to insert heavy objects if they attend to the functional properties of the task or if they can quickly form an association between the more functional option and success. However, unlike in most previous experiments (e.g., *Cheke, Bird & Clayton, 2011*; *Taylor et al., 2011*; *Jelbert et al., 2014*; *Logan et al., 2014*, but see *Logan et al., 2016*), the light objects sank rather than floated, thus if enough were inserted, the food could be reached. I made this modification so that in Experiment 4 (Heavy vs. Light Magic) when the heavy objects became non-functional by sticking to a magnet placed inside the tube above the water, the light objects would now be the functional option because they could fall past the magnet into the water. Individuals that prefer heavy objects or have no preference in the Heavy vs. Light experiment should change their preference in the Heavy vs. Light Magic experiment to preferring neither object or light objects. This would indicate that their preferences are sensitive to changing contexts.

In the Heavy vs. Light experiment, one standard water tube was presented with 4 heavy (steel rod wrapped in fimo clay, weight = 10 g, each displaces 2–3 mm of water) and 4 light (plastic tube partially filled with fimo clay, weight = 2 g, each displaces 1–1.5 mm of water) objects placed in pseudorandomized (as explained for the color association test) pairs near the top of the tube (both objects were 21–24 mm long and 8 mm in diameter; Fig. 5A). Heavy objects had a larger volume (1,056–1,207 mm$^3$) and displaced 0.5–2 mm more water than light objects (volume roughly 500 mm$^3$), which had a hollow end. Thus the heavy objects were more functional than the light objects, but importantly, both objects were functional. Each bird had three opportunities to interact with the objects before the experiment began: one heavy and one light object was placed on the table (pseudorandomized for side) with food underneath and on top of each object. The object that was first touched was recorded and a trial continued until the bird interacted with both objects. If one object was preferred (as indicated by approaching it first 2–3 times), then more food was placed on the other object to try to eliminate any object preference before the experiment began. Each grackle needed to choose each object type first at least once during this process, which resulted in Horchata receiving four interactions and Batido five to ensure a lack of preference. After object interaction trials, each bird was given the 20-trial experiment.

*Experiment 3: results*

Grackles varied in how efficiently they solved problems. Four grackles (Tequila, Margarita, Batido, and Refresco) were 3.4–5.2 times more likely to choose heavy objects rather than the less functional light objects, while two grackles (Cerveza and Horchata) had no preference (they were 0.6–1.4 times more likely to choose heavy objects; see Table 2 for binomial test results and Table 3 for GLMM results). Cerveza and Horchata's performances improved

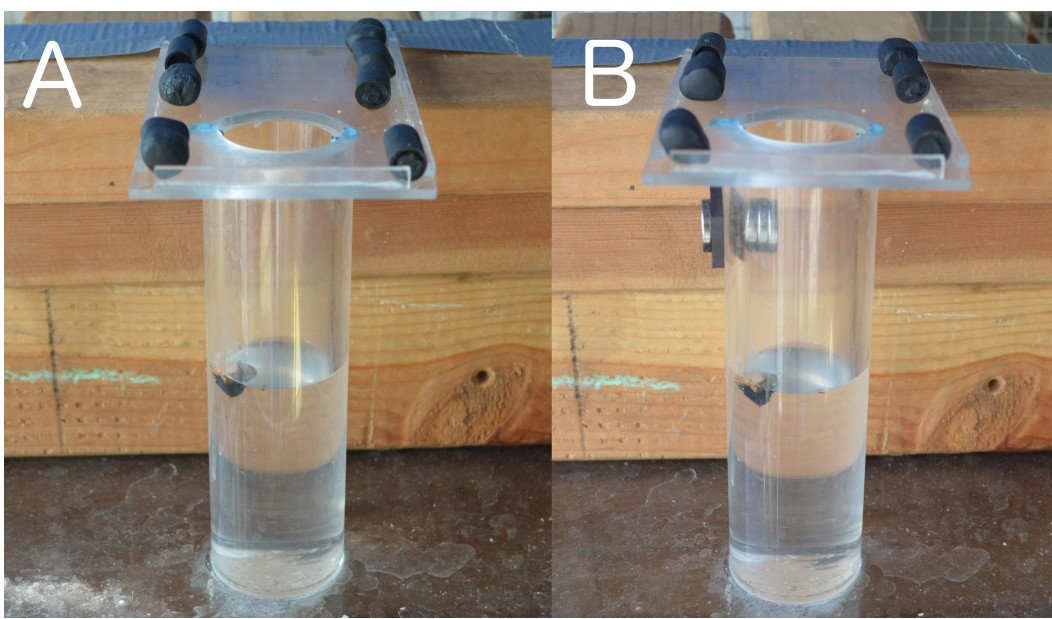

**Figure 5** **Heavy vs. Light (A) and Heavy vs. Light Magic (B).** Heavy objects are more functional than light objects in the Heavy vs. Light experiment (A), while the light objects are the only functional objects in the Heavy vs. Light Magic experiment (B).

across trials: they were 3.9–4.4 times more likely to choose heavy objects as trial number increased, indicating that they learned through trial and error that the heavy objects were more functional (Table 3). The other grackles' performances did not improve with increasing trial number, indicating that they might have been biased toward heavy objects from the beginning of the experiment or perhaps they used prior knowledge to solve the task (e.g., causal knowledge about functional differences regarding weight or volume; Table 3). Horchata was not motivated to participate in the water tube experiments: she required bait between almost all trials to get her to continue to interact with the apparatus, which might have influenced her lack of success. All choices in all trials for all birds are presented Table S1.

### Experiment 3: discussion

Despite not being a tool-using species, grackles performed well in the tool-using Aesop's Fable paradigm object discrimination tests. Four out of 6 grackles discriminated between object types as indicated by their preference for inserting heavy objects significantly more than light objects. Their object discrimination performance is similar to that in other successful species where individuals preferred to insert heavy objects that sank rather than light objects that floated and thus were not functional at all (*Cheke, Bird & Clayton, 2011*; *Cheke, Loissel & Clayton, 2012*; *Taylor et al., 2011*; *Jelbert et al., 2014*; *Logan et al., 2014*). This is in contrast to 4-year-old children who performed poorly by having no object preference (*Cheke, Loissel & Clayton, 2012*) and Western scrub-jays who successfully obtained the food but did not discriminate between object types (*Logan et al., 2016*). Perhaps individuals who discriminated between object types did so because they

**Table 3** **Examining test performance and learning effects in the water tube experiments (3–6).** Examining the influence of experiment, trial, and bird on test success (Test Performance) and whether success increased with trial number (Learning Effects), thus indicating a learning effect.

| | Test performance | | | Learning effects | | |
|---|---|---|---|---|---|---|
| | Posterior mean | Lower 95% CI | Upper 95% CI | Posterior mean | Lower 95% CI | Upper 95% CI |
| Choice number | 0.002 | 1.39E−16 | 0.002 | – | – | – |
| *Heavy vs. light* | | | | | | |
| *Batido* | *1.36* | *−0.03* | *2.83* | −0.01 | −0.13 | 0.12 |
| Margarita | 0.29 | −2.31 | 2.83 | 0.04 | −0.17 | 0.26 |
| Cerveza | −1.92 | −4.06 | 0.10 | 0.13 | 0.05 | 0.32 |
| Horchata | −1.01 | −3.14 | 1.14 | 0.01 | −0.20 | 0.19 |
| Refresco | −0.13 | −2.07 | 2.13 | −0.01 | −0.19 | 0.16 |
| Tequila | 0.22 | −1.96 | 2.47 | −0.003 | −0.20 | 0.22 |
| *Heavy vs. light magic* | | | | | | |
| Batido | −2.11 | −4.66 | −0.11 | −0.06 | −0.25 | 0.16 |
| Margarita | −0.20 | −3.63 | 3.14 | −0.01 | −0.32 | 0.28 |
| Cerveza | 2.13 | −1.33 | 4.86 | −0.18 | −0.45 | 0.15 |
| Horchata | 2.60 | −0.83 | 5.87 | −0.03 | −0.31 | 0.28 |
| Refresco | −1.32 | −4.73 | 1.92 | 0.33 | 0.03 | 0.62 |
| Tequila | −1.05 | −4.50 | 2.43 | 0.17 | −0.17 | 0.50 |
| *Narrow vs. wide* | | | | | | |
| Batido | −1.12 | −3.10 | 0.88 | 0.01 | −0.15 | 0.19 |
| Cerveza | −0.21 | −3.61 | 2.74 | 0.06 | −0.20 | 0.33 |
| Refresco | −0.48 | −3.49 | 2.77 | 0.02 | −0.23 | 0.25 |

**Notes.**

GLMM: Choices Correct ~Experiment*Trial*Bird, random, ~Choice Number. CI, credible intervals, Batido in italics indicates the intercept.

discriminated between the causal properties of the objects, and thus used causal cognition to solve this task. However, other explanations cannot be ruled out yet: they may have had an initial preference for heavy objects (object bias hypothesis), they might have noticed that inserting a heavy object brings the food closer to the top of the tube than inserting a light object (perceptual-motor feedback hypothesis), or they may have associated retrieving food with the heavy objects (*Jelbert, Taylor & Gray, 2015*).

To begin to address the object bias hypothesis, grackles had a modified version of Heavy vs. Light where the light objects, rather than floating and being non-functional, displaced about half the amount of water as the heavy objects (as in *Logan et al., 2016*). In previous studies testing sinking vs. floating, all participating individuals preferred the sinking objects: 2/2 Eurasian jays (*Cheke, Bird & Clayton, 2011*), 4/4 New Caledonian crows (*Taylor et al., 2011*), 6/6 New Caledonian crows (*Jelbert et al., 2014*), 6/6 New Caledonian crows (*Logan et al., 2014*), and children age 5 and over (*Cheke, Loissel & Clayton, 2012*). Additionally, in a similar Aesop's Fable experiment using objects of the same weight but different volumes (Solid vs. Hollow), all individuals chose the larger volume significantly more: 5/5 New Caledonian crows (*Jelbert et al., 2014*) and 6/6 New Caledonian crows (*Logan et al., 2014*). This led to the alternative hypothesis that these individuals, rather than attending to function, had a bias toward heavy objects because they were more similar to familiar objects in the

wild than the light objects (*Logan et al., 2014*; *Jelbert, Taylor & Gray, 2015*). Making both objects functional, but to different degrees, allowed me to partially test the object bias hypothesis: rather than making a dichotomous choice where the heavy object was the only functional option, birds could either have an object bias and/or attend more closely to the functional differences in the properties of the objects and choose the heavy object more, or exhibit no object bias by having no object preference and still succeed. Indeed, Western scrub-jays that participated in this modified Heavy vs. Light experiment showed no object preferences, indicating that birds might not have a general bias toward heavy objects (*Logan et al., 2016*). Two grackles had no object preference and successfully retrieved the reward, which further supports the notion that, at the species level, object biases are not the default. The object bias hypothesis cannot be ruled out for the 4 grackles that preferred to insert heavy objects when both objects were functional. The heavy objects were approximately the same weight as the stones used in stone dropping training, therefore individuals that preferred heavy objects may have relied on an association between weight and function instead of, or in addition to, any similarity the heavy objects might have had to objects in the wild.

Three of the 4 grackles that preferred heavy objects did not show a learning effect across the 20 trials in this experiment, indicating that they relied on prior information about the world to solve this task, which suggests that they had an object bias or they may have used causal cognition. Their performance suggests that they did not simply associate the heavy objects with reaching the food because both object types could result in a reward.

## Experiment 4: heavy vs. light magic (behavioral flexibility)
### Experiment 4: methods
The set up was the same as in Experiment 3, except there were magnets (2 super magnets on the outside and 3 on the inside of the tube) attached to the tube above the water level such that the heavy objects would stick to the magnets and not displace water, while the light objects could fall past the magnets into the water, thus being the functional choice (Fig. 5B). Birds were given 3 heavy and 3 light objects, placed in pseudorandomized pairs near the top of the tube, and 20 trials were conducted.

### Experiment 4: results
Two grackles exhibited behavioral flexibility by changing their preference from Experiment 3. Tequila and Refresco changed from choosing significantly more heavy objects in Experiment 3 to having no object preferences in this experiment (binomial test: $p = 0.60$ and 1.00, respectively), whereas Batido continued to prefer the non-functional heavy objects (binomial test: $p = 0.02$) and Horchata continued to have no object preference (binomial test: $p = 1.00$; see Table 2 for binomial test results, Table 3 for GLMM results, and Table S2 for all choices made by all birds). Margarita continued to prefer heavy items (binomial test: $p = 0.02$) and Cerveza went from having no preference to choosing significantly more non-functional heavy items (binomial test: $p = 0.02$) because they exhibited an intense interest in the magnet (Table 2; see a video clip at: https://youtu.be/fdCJGwvaDsk). They repeatedly stuck heavy objects to the magnet and attempted to pull them off, requiring almost no rewards between trials for participating, which indicated a high degree of motivation

(motivation that rapidly decreases if they fail experiments). The experiment did not have the intended effect on their behavior. Tequila gave up after 17 trials, refusing to drop either type of object into the tube, indicating he may have inhibited his heavy preference while at the same time not persisting with the light objects. Tequila and Refresco's performance improved with trial number, indicating that they learned through trial and error about which object was functional (Table 3). The other grackle's performances did not change or they decreased with increasing trial number, indicating that they did not learn about which object was functional (Table 3). Even though Tequila and Refresco did not learn to prefer light objects in the amount of trials given, they did exhibit flexibility in that they changed their preferences from heavy in the previous experiment to having no preference in this experiment.

### Experiment 4: discussion

Behavioral flexibility was exhibited by grackles because they changed their preferences when the task changed. When the heavy objects in the Heavy vs. Light Magic experiment were no longer functional because they stuck to a magnet, 2 grackles changed from having preferred heavy objects when they were functional in Heavy vs. Light to having no object preference in the Magic experiment. This demonstrates either an attention to the functional properties of objects or that some grackles who previously might have been biased toward heavy objects were able to counteract this bias in light of new associative information (i.e., associating light objects with the previously rewarded movement of an object falling into the water). Either way, their behavior changed when circumstances changed.

No grackle completely switched their preference to the light objects, which may have been due to the difficult design of the apparatus, both theoretically and physically. Theoretically, this experiment is similar to the U-tube Aesop's Fable experiment where birds that rely on causal cues to solve these kinds of tasks should fail because no causal information is available about how the apparatus works (Cheke, Bird & Clayton, 2011; Jelbert et al., 2014; Logan et al., 2014). Instead, birds must rely solely on associative cues to solve the task (e.g., dropping stones into the blue tube results in a reward in its adjacent tube). If birds generally solve these kinds of problems associatively, then they should learn this association at a similar rate to other Aesop's Fable experiments. However, if birds rely to some degree on causal cues to solve these problems, then their learning speed should be impeded. The grackle's lack of learning to associate light objects with a food reward may have been partially due to an attendance to causal cues when solving these tasks. The apparatus was difficult physically as well: if one heavy item was inserted into the tube, it stuck to the magnet and blocked access to the food regardless of how many light objects were inserted. Thus, grackles had to inhibit inserting any heavy objects to solve this problem, which made the task difficult. Tequila and Refresco's performance is consistent with previous interpretations of Eurasian jay behavior on the U-tube apparatus: they were slow to learn from associative cues when only associative information was available, therefore they likely rely to a degree on causal cues when solving these tasks (Cheke, Bird & Clayton, 2011). However, in the case of the grackles, I cannot rule out that the physical difficulty of the apparatus was the cause of

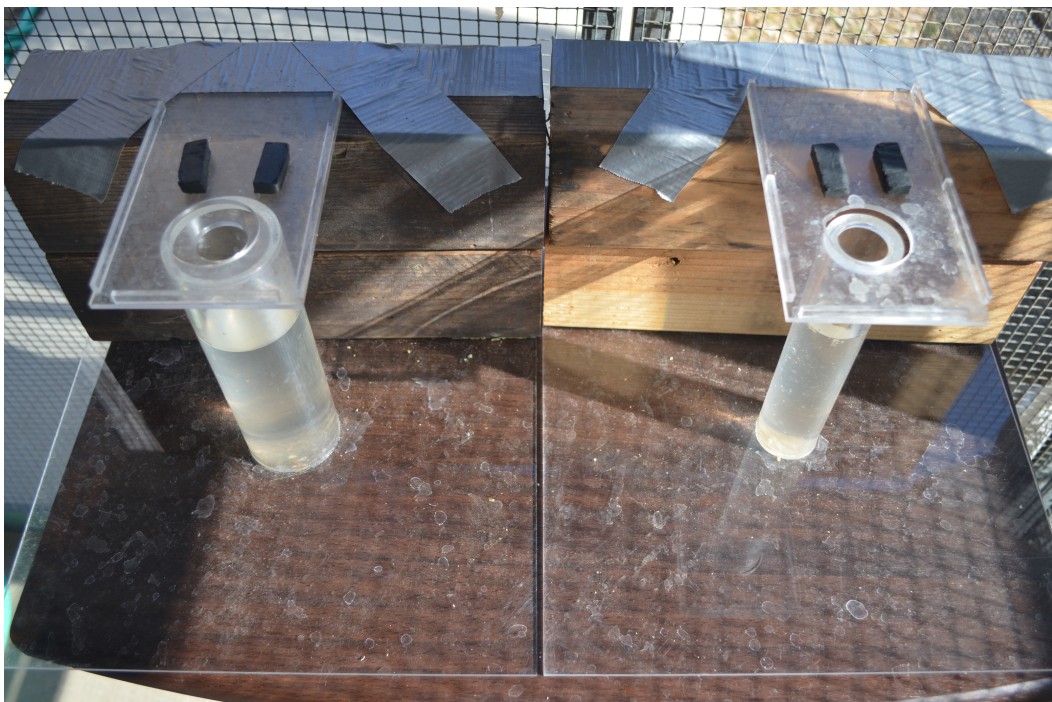

**Figure 6** **Narrow vs. Wide.** Dropping clay objects into the narrow tube in the Narrow vs. Wide equal water level experiment is the only way to reach the floating food.

their slow learning, which, if addressed, might have resulted in their quickly learning this task using associative cues.

## Experiment 5: narrow vs. wide equal water levels (problem solving ability)

### Experiment 5: methods

Experiments 5 and 6 followed the same methods used for New Caledonian crows (*Logan et al., 2014*). To solve Experiment 5, objects must be inserted into a narrow (functional) rather than a wide (non-functional) tube when water levels are equal in both tubes. In Experiment 6, the narrow tube becomes non-functional because the water level is too low, therefore birds must change their preference to inserting objects into the functional wide tube or to having no preference (as long as they successfully reach the food in most trials) to demonstrate behavioral flexibility.

To determine whether birds attend to volume differences in Experiment 5, a wide tube and a narrow tube with equal water levels were presented with four objects made out of fimo clay (30 × 10 × 5 mm, 3–4 g, each object displaced 1–2 mm in the wide tube and 5–6 mm in the narrow tube; *Logan et al., 2014*; Fig. 6). Two objects were placed on the platform near the narrow tube opening and two objects on the platform near the wide tube opening. The objects were functional only if dropped into the narrow tube because the water levels were set such that dropping all of the objects into the wide tube would not bring the floating food within reach. However, dropping 1–2 objects into the narrow tube would raise the water level enough to reach the food. Both tubes were 170 mm tall with 3 mm thick lids

that constricted the opening to 25 mm in diameter to equalize the bird's access to the inside of each tube, and super glued to a clear acrylic base (300 × 300 × 3 mm). The wide tube (outer diameter = 57 mm, inner diameter = 48 mm, volume = 307,625 mm$^3$) was roughly equally larger than the standard water tube (dimensions above, volume = 192,800 mm$^3$) as the narrow tube was smaller (outer diameter = 38 mm, inner diameter = 25 mm, volume = 83,449 mm$^3$). The position of the tubes was pseudorandomized for side to ensure that tube choices were not based on a side bias, and 20 trials were conducted. Before the experiment began, each bird had three opportunities to interact with the object, as in Experiment 3, only here it was simply to habituate them to the clay object (one object type) and not to train the birds not to prefer one object type over another.

### Experiment 5: results

Grackles did not discriminate between water volumes, indicating they were not efficient problem solvers in this context. All three grackles that participated in this experiment displayed no preference for dropping objects into the functional narrow tube or the non-functional wide tube (binomial test: $p = 1.00$ for every bird, Table 2; see Table 3 for GLMM results, and Table S3 for all choices by all birds). None of the grackles' performances improved with trial number, indicating that they did not learn to distinguish which tube was functional (Table 3). Batido appeared to rely on the strategy of dropping all objects into both tubes regardless of which tube he received a reward from, although in trial 12, he picked up the objects from the wide tube area and dropped them into the narrow tube even though he was trained to only push stones, not drop them (Table S3).

Some grackles did not initially transfer from dropping previous object types to dropping the clay objects used in this experiment. It appeared as though they were trying to solve the problem, but did not perceive the clay objects as being the kind of object one would drop into a water tube. In these cases, additional training was implemented using a single standard water tube and a mixture of clay objects and stones until the bird was willing to drop objects into the tube even if they consisted only of clay objects. Cerveza transferred to dropping clay objects after 4 training trials, but Tequila and Margarita were excluded from this experiment because they did not transfer to dropping clay objects into tubes. After 14 training trials on a regular water tube with stones and clay objects available to Tequila, it was clear that it would take many more training trials than there was time for because his motivation was greatly diminished. Margarita refused to participate in the training trials. Horchata was also excluded from this experiment because she refused to interact with the objects.

### Experiment 5: discussion

Grackles did not discriminate between water volumes in the Narrow vs. Wide equal water level experiment. Perhaps their skill in water displacement experiments is limited to objects, however more experiments involving object and tube properties would need to be conducted to confirm this.

### Experiment 6: narrow vs. wide unequal water levels (behavioral flexibility)

***Experiment 6: methods***

Those grackles that passed Experiment 5 continued to this experiment to determine whether their tube choices adjusted to changing circumstances. This experiment was the same as Experiment 5, except the water level in the narrow tube was lowered to 5 cm from the table, thus making the food unreachable even if all objects were dropped into this tube (as in *Logan et al., 2014*). The water level in the wide tube was raised such that the bird could reach the food in 1–2 object drops, and 20 trials were conducted.

***Experiment 6: results***

No grackle passed Experiment 5, indicating they were not sensitive to the differences in water volumes; therefore they were not given Experiment 6, which would have investigated their behavioral flexibility in this context.

***Experiment 6: discussion***

While grackles were not given Experiment 6, it is interesting to compare their behavioral flexibility in the Heavy vs. Light experiments with New Caledonian crow performances in the Narrow vs. Wide experiments because crows previously showed behavioral flexibility in this context (New Caledonian crows were not given the Heavy vs. Light Magic experiment because it was not invented yet, which is why I cannot directly compare their behavioral flexibility performances). Four out of 6 New Caledonian crows preferred to drop objects into the functional narrow tube rather than the non-functional wide tube. When the wide tube became the functional option, 3 New Caledonian crows changed their preference to the wide tube and 1 changed to no preference (*Logan et al., 2014*). Grackle performances in the Heavy vs. Light experiments were similar to the New Caledonian crow that changed from narrow to no preference in the Narrow vs. Wide experiments.

## Context 2: first choices on first trials

First trial performance on tasks that can involve more than trial and error learning is considered an important feature when evaluating cognitive performance because it gives an indication of what the individuals might infer about the task. All six grackles chose the more functional heavy objects as their first choice in their first trial in Heavy vs. Light, which indicates that they preferred the heavy objects from the very beginning of the experiment (Table S1). Five out of six grackles chose the non-functional heavy objects in Heavy vs. Light Magic (Table S2), which is not surprising given that they had learned to prefer heavy objects in the previous experiment and had likely never interacted with a magnet before, therefore they should have had no reason to have a prior understanding of how the Magic experiment worked. Two out of three grackles chose the functional narrow tube in Narrow vs. Wide with equal water levels, indicating no group-level initial preference for a particular tube (Table S3).

## Context 2: Did choice number influence the results?

Individuals could learn how the water tube tasks worked with each choice they made, potentially making each choice dependent on previous choices. Multiple choices could be

**Table 4 Summary of results.** The learning strategy or strategies employed by each bird as well as the number of trials to reach proficiency in Experiment 1, the number of trials needed to reverse their preference (Experiment 2) minus the number of trials needed to initially learn the preference (Experiment 1; a measure of behavioral flexibility), and whether they exhibited a preference change between Experiments 3 and 4. (No) = preferred to stick heavy objects to the magnet thus the experiment did not test what it was designed to test in these individuals, X = did not participate in this experiment. — = did not complete this experiment.

| Bird | Sex | Learning strategy | Experiment 1: Learning speed | Behavioral flexibility (Exp 2–Exp 1) | Behavioral flexibility (water tube) |
|---|---|---|---|---|---|
| Tequila | M | Epsilon-decreasing | 30 | 70 | Yes |
| Margarita | F | Epsilon-decreasing | 30 | 70 | (No) |
| Cerveza | F | Epsilon-decreasing | 30 | 60 | (No) |
| Horchata | F | Epsilon-decreasing | 30 | 100 | No |
| Refresco | M | Epsilon-first, then epsilon-decreasing | 20 | 50 | Yes |
| Batido | M | Epsilon-decreasing | 30 | – | No |
| Michelada | F | Epsilon-decreasing | 40 | 30 | X |
| Jugo | M | Epsilon-decreasing | 40 | 40 | X |

made per trial; therefore I analyzed how independent choice number was. Choice number was modeled as a random factor in the GLMM and did not influence the results, indicating that choices appear independent of each other (Table 3). This result combined with the result that some individuals increased their success across the 20-trials, indicates that learning occurred at the trial level, not at the level of individual choices. Perhaps learning was linked to the receipt of a food reward, which was obtained near the end of a trial, rather than the movement of the food in the tube, which could occur with each choice.

### Did behavioral flexibility correlate across contexts?

Those grackles that were more behaviorally flexible in the Aesop's Fable context (i.e., changed preferences between Experiments 3 and 4) were not more behaviorally flexible in the color association context (i.e., faster to reverse a previously learned preference (Experiments 1 and 2); Spearman's rank correlation test: $S = 28.89$, $p = 0.45$, rho $= -0.44$, $n = 5$; Table 4).

## GENERAL DISCUSSION

Grackles demonstrated behavioral flexibility in two contexts, the Aesop's Fable paradigm and a color association test. Contrary to predictions, behavioral flexibility did not correlate across contexts: Those grackles that exhibited flexibility in the Aesop's Fable paradigm were not the fastest to reverse a previously learned color preference in the color association test. Four out of 6 grackles exhibited efficient problem solving abilities, similar to other successful species, by choosing heavy objects significantly more in the Heavy vs. Light experiment. Problem solving efficiency did not appear to be directly linked with behavioral flexibility because not all grackles (2/4) that were efficient problem solvers changed their preference in the Aesop's Fable paradigm. Problem solving speed (Experiment 1) also did

not significantly correlate with reversal learning scores (Experiment 2), indicating that faster learners were not the least flexible.

The lack of correlations within and across contexts for behavioral flexibility and for behavioral flexibility and problem solving supports the hypothesis that behavioral flexibility is not only an independent source of variation distinct from problem solving ability and speed (*Cole, Cram & Quinn, 2011*), but also independent across contexts (*Griffin & Guez, 2014*). This finding suggests that investigations of behavioral flexibility should conduct multiple tests of behavioral flexibility in different contexts to more fully understand how it relates to itself, cognition, and other variables of interest (*Griffin & Guez, 2014*). Maintaining many independent sources of variation in a population could be useful for successfully adapting to new environments. Western bluebirds were found to rely on a range of dispersal strategies that already existed in their population when colonizing a new habitat (*Duckworth, 2008*). Thus, the more sources of variation a population can access, the more likely it is that at least some of this variation will suit the new situation and allow the species to adapt.

One potential explanation for why individuals varied in behavioral flexibility across contexts regards the type of cognition used in each context. Causal cognition and/or trial and error learning based on multiple cues (e.g., object type, movement of the food with each object drop) could be used to solve the Aesop's Fable tasks (Experiments 3–6), whereas only trial and error learning based on one cue (i.e., color) could be used to solve the color association tasks (Experiments 1–2). If individuals varied in the kinds of cues they attended to this might have caused the differences in performance across contexts. Such individual variation in attention to particular cues was found in Eurasian jays (*Cheke, Bird & Clayton, 2011*). Behavioral flexibility will need to be tested in more contexts to determine whether individual differences are due to differential cue use or different contexts.

A higher number of learning strategies in the color association tests did not necessarily indicate flexibility in the color association context or the Aesop's Fable context. Refresco was one of the two behaviorally flexible individuals in the water tube experiments (3–4), and about average in reversing a color preference (Experiment 2; Table 4). He was also the only grackle to use more than one learning strategy in the color association experiments: he used the epsilon-first strategy to sample the environment once before arriving at the correct solution and then he stayed with the correct choice for the rest of Experiment 1. He then switched his learning strategy to epsilon-decreasing for his color association refresher and for reversal learning (Experiment 2), which is the same strategy the rest of the birds used in Experiments 1 and 2. Individuals using the epsilon-decreasing strategy sample the environment extensively before consistently making the correct choice. Because there was almost no individual variation in learning strategies it is difficult to understand how this trait covaries with behavioral flexibility. However, it suggests that a variety of learning strategies is not required for a large amount of variation in behavioral flexibility, problem solving ability, and problem solving speed to exist in a population.

That behavioral flexibility did not correlate across contexts or with problem solving ability (Experiment 3) or speed (Experiment 1) reveals how little we know about behavioral

flexibility, and provides an immense opportunity for future research to explore how individuals and species can use behavior to react to changing environments.

## ACKNOWLEDGEMENTS

I am grateful to Luisa Bergeron, Christin Palmstrom, Linnea Palmstrom, and Michelle Gertsvolf for trapping and aviary assistance; Alexis Breen for helping to train Jugo; Brigit Harvey for stone dropping training consultations; Steve Rothstein for scouting grackles and for use of the aviaries; Joe Jablonski and David Bothman for making the apparatuses; Jill Zachary and Kathy Frye at Santa Barbara City Parks and Recreation for use of the Andree Clark Bird Refuge and East Beach Park; Estelle Sandhaus and Chris Briggs at the Santa Barbara Zoo for access to wild grackles; Karrie Black for managing purchasing and grants; Alex Thornton for logistical advice; Dieter Lukas for conceptual input and analysis feedback, Alecia Carter for manuscript restructuring advice, and the rest of LARG for feedback; Krista Fahy at the Santa Barbara Museum of Natural History for helping with permit applications; Kristine Johnson, Sarah Overington, Julie Morand-Ferron, and Neeltje Boogert for grackle advice and trap plans; Manny Garcia for veterinary and permit support; Mary Hunsicker and Bertrand Lemasson for assistance with making the trap; Margaret Tarampi, Eric Egenolf, Rebecca Schaefer, and Sam Franklin for brainstorming object designs; Will Hoppitt for GLMM effect size interpretation assistance; and Irina Mikhalevich, Ljerka Ostojić, Neeltje Boogert, Jennifer Vonk, Zoe Johnson-Ulrich, and an anonymous reviewer for manuscript feedback.

### Funding

Funding was provided by the National Geographic Society/Waitt Grants Program (grant number W252-12) and Junior Research Fellowship from the SAGE Center for the Study of the Mind at the University of California Santa Barbara. The funders had no role in study design, data collection and analysis, decision to publish, or preparation of the manuscript.

### Grant Disclosures

The following grant information was disclosed by the author:
National Geographic Society/Waitt Grants Program: W252-12.
Junior Research Fellowship from the SAGE Center for the Study of the Mind at the University of California Santa Barbara.

### Competing Interests

The author declares there are no competing interests.

### Author Contributions

- Corina J. Logan conceived and designed the experiments, performed the experiments, analyzed the data, contributed reagents/materials/analysis tools, wrote the paper, prepared figures and/or tables, reviewed drafts of the paper.

**Animal Ethics**

The following information was supplied relating to ethical approvals (i.e., approving body and any reference numbers):

This research was carried out in accordance with permits from the US Fish and Wildlife Service (scientific collecting permit number MB76700A-0), California Department of Fish and Wildlife (scientific collecting permit number SC-12306), US Geological Survey Bird Banding Laboratory (federal bird banding permit number 23872), and the Institutional Animal Care and Use Committee at the University of California Santa Barbara (IACUC protocol numbers 860 and 860.1).

**Data Availability**

Logan C. 2016. Great-tailed grackle behavioral flexibility and problem solving experiments, Santa Barbara, CA USA 2014–2015. KNB Data Repository. https://knb.ecoinformatics.org/#view/doi:10.5063/F1319SVV.

**Supplemental Information**

Supplemental information for this article can be found online at http://dx.doi.org/10.7717/peerj.1975#supplemental-information.

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
