# Peer review of "Behavioral flexibility and problem solving in an invasive bird"

_PeerJ, doi:10.7717/peerj.1975_

## Round 0.1 · original submission · Major Revisions

Two expert reviewers and I have now had a chance to review your MS. Both reviewers were enthusiastic about the experimental approach but had some reservations about your conclusions. I’d also like to applaud you for an extensive series of studies to investigate a problem that has not received a huge amount of attention, particularly in this species. I’d like to recommend that you revise the MS with a more cautious interpretation of the results following the reviewers’ recommendations.

Reviewer 1 included an annotated PDF with comments. Please be sure to attend to the comments on this annotated version in addition to the reviews themselves when preparing your revision.

I have some additional comments of my own:

There is some theoretical discussion of the trade-off between inhibition and innovation that might be relevant to your discussion of behavioral flexibility in relation to problem solving speed in the opening paragraphs.

I do not share Reviewer 1’s concern with the term “preferences” in general, but in this context, if you are referring to a trained behavioral discrimination rather than a spontaneous preference, the reviewer is correct that “preference” may be misleading.

More detail is needed with regard to the learning strategy discussion on pg. 2. (lines 41-45 and 271-287). Readers will need more help to understand the analysis applied here.

A reference is needed on line 53.

I’d prefer the avoidance of value-laden terms such as “good” problem solvers. Stick to more objective terms like ‘quick’ or ‘accurate’ depending on the measure used to determine problem solving ability.

The modified Aesop’s fable task is clever, but I would urge you to consider seriously the fact that eliminating the ability of the heavy object to sink by using magnets would work against any causal attributions the grackles might make about the role of weight in the task. There is no opportunity for them to understand the role of the magnet and its unique effects on the heavy object so, if the grackles were reasoning causally rather than associatively, this task might be slower to learn rather than faster. Here it is important to consider how flexibility may not be correlated with abilities such as causal reasoning. Thus, we should be cautious in how much weight we give each trait in contributing to underlying “intelligence” or “problem—solving ability”.

On lines 109-111, you write “If habituation to an apparatus was needed..” How was this determined? Did this happen every time an apparatus was novel? If so, state as such. Otherwise it sounds like it was determined arbitrarily. Similarly, when was re-habituation needed? (line 113).

After how long a period of inactivity were trials terminated (lines 123-124)?

On lines 207-208 you indicate that more food was placed on the non-preferred object to try to eliminate a lack of preference. Did you objectively ensure unbiased responding before moving on to the next experimental phase?

On line 225, change “The objects were only functional” to “The objects were functional only if…” On line 269, change “only allowed” to “allowed only”. Please make similar changes throughout as necessary.

No mention is made of filming the trials for reliability purposes. This is essential to ensure unbiased reporting of the results. Ideally, a coder naïve to the hypotheses of the study should code at least a portion of the trials.

Lines 288-293 imply that a one-tailed test was used. Please clarify. You should be using two-tailed tests because it is possible that the grackles could perform below chance.

Early on you indicate that all grackles participated in the experiments in the same order but on line 371 you indicate that the order was different for Tequila. Please be consistent.

You write about changing preferences in Experiment 3 but there is no description of a statistical analysis to support this contention – just a reference to tables. Please insert some description of the analyses conducted and the results in the text itself.

Experiment 5 should not be discussed if it was never implemented.

I find it a bit difficult to follow the key findings as I progress through the Results section. I think a clearer statement of the tested hypotheses, aligned with the statistical or graphical approach to assessing these outcomes and a summary based on the performance of the birds in general would be helpful for each experiment. Generally I prefer a methods/results/discussion section for each experiment in turn but I recognize you may have been conforming to standard practice for this journal.

The discussion reads a bit like a laundry list of findings with a few comments on each finding, and then moves on to the next finding. Please work on synthesizing the results in line with previous research and the key hypotheses for the study.

Did you look at how performance was correlated across tasks? Your statement on lines 560-561 is not backed up with evidence.

The conclusion could be strengthened by discussing variability in behavioral flexibility in more detail. What does it mean if only a few grackles from your modest sample displayed this trait? What does it mean that it is not stable across contexts?

There are a lot of tables and figures. I am not sure that all are absolutely necessary. Perhaps tables 4-7 could be supplemental.

Thank you for submitting such interesting work to PeerJ.

·

Basic reporting

The article adheres to PeerJ polices and uses clear and unambiguous text. I personally dislike the use of the term “preference” to indicate behavioral responses during the experiments, especially as many of the “preferences” are learned during the experiments. I feel a better term would be “response” or “behavior” as "preference" has emotional connotations and is less direct when you are actually referring to learning and responses the subjects make in experimental contexts.

The structure conforms the templates. Figures and tables are relevant and useful. The submission is self-contained and a good unit of publication; the inclusion of 6 related experiments helps the article make a strong argument.

The introduction could include more background information on behavioral flexibility. The author makes argument that mechanics of behavioral flexibility are not well understood (I agree) but the author does not cite enough of the literature on behavioral flexibility to thoroughly demonstrate this; the author should discuss the many contexts in which behavioral flexibility has been studied. In addition, all experiments revolve around reversal learning as the measure of behavioral flexibility so the introduction would be greatly enhanced by an expanded discussion of reversal learning, how reversal learning is a good measure of behavioral flexibility, and previous work on reversal learning. I’ve included notes in the manuscript of areas that could be expanded or more literature cited.

Experimental design

The submission describes original, primary research within the scope of the journal. The submission clearly defines the research question, which is important and meaningful. The author aims to investigate the mechanics of behavioral flexibility (measured by reversal learning) in two contexts (learning speed and problem-solving). The author identifies this knowledge gap and states that this will contribute to filling that gap. As discussed in “Basic Reporting” the author could more strongly present this as a meaningful gap with inclusion of references to more literature on behavioral flexibility (I think this literature can easily be included and that the research question will still be relevant and meaningful with the inclusion of this literature). Some literature suggestions are referenced in my comments on the manuscript.
The investigation overall was conducted rigorously and to a high technical standard; methods are sufficiently described and conducted ethically.

Validity of the findings

The data is robust, statistically sound, and controlled; full dataset is available online. The use of an economic analysis to investigate learning strategies is particularly impressive and not often included in studies on problem-solving; this is a welcome addition that strengthens the article.

The conclusions are appropriately stated and connected to the original question, however the author is a bit strong in attributing the potential for causal cognition by the subjects (see specific comments on the manuscript). The author suggests that because the subjects discriminate between objects on one property (weight) that this could indicate the use of causal cognition by the subjects. I agree that this potential exists, but would prefer the author to state this more cautiously. The ability to discriminate on a property that happens to be functional does not indicate causal cognition, although it is certainly a precursor for causal cognition. The use of causal cognition can only be demonstrated if the subjects understand the functional property (i.e. know why the functional property is relevant or have a concept of weight as an unobservable causal relationship between objects and water displacement). If the subjects used weight cues in functional contexts, but not non-functional contexts, this would be a stronger indication of causal cognition than just discriminating on a property that happens to be functional. The author speaks about causal cognition broadly without indicating that weight is the functional property of interest; indeed a discussion of the causal reasoning literature on weight would be welcome if the author wishes to discuss the possibility of the subject’s using causal cognition about weight.
The author does identify the attributions of causal cognition as speculation but appears to find this the most credible hypothesis, which is overstated.
Other than the discussion of causal cognition, the results, discussion, and conclusion are well stated and strong.

Reviewer 2 ·

Basic reporting

Lines 46-75 seem out of place in the introduction; incorporating these lines into the methods and discussion sections may increase readability and reduce redundancy.

Experimental design

The description of the learning strategies is unclear and could use more explanation on how epsilon was measured in the context of these experiments.

line 221: "To determine whether birds understand volume differences". The word understanding implies a level of cognitive awareness/complexity that is not being tested here. The words "attend to" may be more appropriate.

Lines 372 & 375 mention a 'sand filled' tube experiment, but this experiment does not appear in the methods section.

Validity of the findings

The study would benefit from calculating a measure of neophobia for the birds in each task rather than using undefined anthropomorphic terms, see the below instances:
line 351: "Michelada was scared", what was the measure of "scared"? neophobic may be a better term but still requires a measurable definition
line 373: "started to give up" what is the functional definition of give up?
line 120: "If a bird started to lose motivation", what was the measure of motivation?
line 384: "losing motivation"
For example, instead the author could calculate the number of times a subject makes physical contact with the testing apparatus or the amount of trial time a subject spends within a certain distance from the testing apparatus.

line 396: "used prior knowledge". It is unclear how this result indicates the use of prior knowledge.

For the water tube tasks, did the grackles reliably solve the tasks?
The study would benefit from including a measure of success calculated as whether the grackles solved each task above chance levels.

Additional comments

Overall interesting results and a good contribution. However, the author over states the grackles performance on the water tube tasks. The author's main focus is the water tube experiments, however, the results of these experiments are questionable and the author's interpretations highly speculative. The manuscript would benefit from a reduction in water tube speculation and a greater focus on the color task and grackles ability to discriminate color and reverse their color preference.

From the results presented, it does not appear that the "grackles performed well" on the heavy vs. light task. It is unclear whether the grackles discriminated between the functional properties of the 'heavy' and 'light' objects. This is speculative given the results of the 'heavy light magic' task. If grackles attended to the functional aspects of the task, then one would expect them to reverse their preference in the 'heavy light magic' task. However, the majority of grackles did not reduce their preference.
The interpretation that a 'finer degree of object discrimination' is evidenced by grackles preference for heavy objects in the Heavy vs. Light task is not supported by the results. The grackles preference for the heavy object across all tasks is more indicative of an 'innate' preference for heavy objects and grackles previous positive reinforcement with heavy objects and negative reinforcement with the light objects. There is no evidence that the grackles associated the light object with a 'food reward', thus both objects resulting in a food reward is irrelevant given grackles prior experience with the heavy and light objects. The majority of grackles did not discriminate between the heavy vs light object; two grackles having no preference is not strong evidence for causal cognition.

---

## Round 0.2 · Minor Revisions

Thank you for being so responsive to the reviewers’ and my comments and for completing the revisions so promptly. The revisions have greatly improved the flow and readability of the MS and it is now much more clear what you were evaluating and why. I have a few remaining comments I’d like you to address before I can formally accept your manuscript. The line numbers I refer to are based on the tracked version of the document. My comments proceed in chronological order rather than focusing on importance.

On line 20 , I’d change “is moved to the opposite option” to “contingencies are altered such that previously correct choices are now unsuccessful”.
On lines 27-28, I don’t think you need, “
which are declining, but not yet listed as vulnerable: BirdLife International 2012)”
On line 46, change “only in one…” to “in only one context” (Apologies for the continued grammatical nit-picking but these are things that have also stuck with me from my own reviews).
On lines 64-65, I’d replace “compare how fast grackles are at learning and reversing preferences compared with other species” with “compare the speed with which grackles learn and reverse preferences…”
Could you provide more detail or an example of the types of patterns that would predict the kinds of learning strategies you are referring to at the end of lines 70-71?
I think you could do away with the sentence, “All grackles underwent stone dropping training to allow them to participate in Aesop’s Fable experiments” on lines 408-409.
I think you should get rid of speculative comments in the results, such as “likely would have further changed their preference” on line 645 and 609-610. Simply explain what their pattern of choices indicates about what they have learned without speculating about what performance would have been that is not measured.
On line 736, change the , between “volumes” and “therefore” to a ;
Change the “while” on line 740 to “Whereas”. Also on line 1017.
Be sure that it is clear what experiments the discussion pertains to on what is pg. 38 of the tracked version.
Change “Discussion” to “General Discussion” on line 989.
On lines 1003-1005, state “This finding…” and change “highlights the fact investigations..” to something that flows better. Perhaps inserting a “that” before “investigations” would help, but this doesn’t seem like a “fact” so much as a suggestion.

---

## Round 0.3 · accepted · Accept

Thank you once again for your extremely prompt revisions and for being so open to suggested changes. I apologize for the mismatched line numbers. That is very odd. In being unsure what experiments were being referred to I was referring to the sections under subheadings on pages 27 and 28 immediately preceding the general discussion. It appears that these are general findings in summary so I think it is probably clear as it is. Therefore, I am happy to accept this version of the MS. Thank you for once again submitting such interesting work to PeerJ.